# A Review on the Control of the Mechanical Properties of Ankle Foot Orthosis for Gait Assistance

**Dimas Adiputra** [1,2,†][iD]**, Nurhazimah Nazmi** [1,†]**, Irfan Bahiuddin** [3,†][iD]**, Ubaidillah Ubaidillah** [4,†]**, Fitrian Imaduddin** [4,†]**, Mohd Azizi Abdul Rahman** [1,*,†]**, Saiful Amri Mazlan** [1,†] **and Hairi Zamzuri** [1]

[1] Advanced Vehicle System Laboratory, Malaysia-Japan International Institute of Technology, Universiti Teknologi Malaysia, Kuala Lumpur 54100, Malaysia; adimas2@live.utm.my (D.A.); nurhazimah2@live.utm.my (N.N.); amri.kl@utm.my (S.A.M.); hairi.kl@utm.my (H.Z.)

[2] Electrical Engineering Department, Institut Teknologi Telkom Surabaya, Surabaya 60234, Indonesia

[3] Vocational School, Universitas Gadjah Mada, Jogjakarta 55281, Indonesia; irfan.bahiuddin@ugm.ac.id

[4] Mechanical Engineering Department, Universitas Sebelas Maret, Surakarta 57126, Indonesia; ubaidillah@uns.ac.id (U.U.); fitrian.imaduddin@mmu.edu.my (F.I.)

\* Correspondence: azizi.kl@utm.my

† These authors contributed equally to this work.

**Abstract:** In the past decade, advanced technologies in robotics have been explored to enhance the rehabilitation of post-stroke patients. Previous works have shown that gait assistance for post-stroke patients can be provided through the use of robotics technology in ancillary equipment, such as Ankle Foot Orthosis (AFO). An AFO is usually used to assist patients with spasticity or foot drop problems. There are several types of AFOs, depending on the flexibility of the joint, such as rigid, flexible rigid, and articulated AFOs. A rigid AFO has a fixed joint, and a flexible rigid AFO has a more flexible joint, while the articulated AFO has a freely rotating ankle joint, where the mechanical properties of the AFO are more controllable compared to the other two types of AFOs. This paper reviews the control of the mechanical properties of existing AFOs for gait assistance in post-stroke patients. Several aspects that affect the control of the mechanical properties of an AFO, such as the controller input, number of gait phases, controller output reference, and controller performance evaluation are discussed and compared. Thus, this paper will be of interest to AFO researchers or developers who would like to design their own AFOs with the most suitable mechanical properties based on their application. The controller input and the number of gait phases are discussed first. Then, the discussion moves forward to the methods of estimating the controller output reference, which is the main focus of this study. Based on the estimation method, the gait control strategies can be classified into subject-oriented estimations and phase-oriented estimations. Finally, suggestions for future studies are addressed, one of which is the application of the adaptive controller output reference to maximize the benefits of the AFO to users.

**Keywords:** Ankle-Foot Orthosis; gait control; gait assistance; bending stiffness; damping stiffness; assistive torque; motion path; non-linear; walking; mechanical properties

---

## 1. Introduction

Stroke (brain attack) is caused by the interruption of blood flow to the brain, leading to damage to the nerves in the brain and disrupting the exchange of information through the normal path between the limbs and the brain. Fortunately, the exchange of information can be redirected to another path through learning, which requires the limb to repeat a certain task [1]. This series of repetitions is known

as training or rehabilitation, where a therapist assists patients in performing the task. The movements of the limb during the training must be uniformly maintained. In other words, different movements, for instance, caused by muscle pain, should be avoided [2]. It is expected that with a more uniform performance of repetitive tasks, the recovery of the patient will be faster. Robotic technologies, such as orthotic devices and exoskeletons, offer intensive, controlled, and monitored post-stroke rehabilitation to help patients to perform tasks [3]. In addition, the pace of robot-assisted training is faster than with just the therapist alone, thereby reducing the therapist's burden [4]. Therefore, robotic technologies have been developed over the past decade for the assistance and enhancement of the training or rehabilitation of post-stroke patients [5].

One daily activity task is locomotion, which defines the way in which humans move from one place to another. The ability of locomotion, such as walking, is crucial for avoiding anti-social behavior and anxiety over the loss of mobility [6]. Therefore, locomotion is one of the primary tasks to be acquired by a post-stroke patient, in addition to hand motions [7] and speech [8]. The basic form of locomotion is walking, besides running, jumping, and so on. The walking gait is a quasiperiodic activity, as depicted in Figure 1. It starts from the initial contact (IC) on the heel. The foot goes down until the toe touches the ground. This is known as the foot flat (FF). From here, the leg moves forward along with the body. At the end of the FF, the foot pushes the ground with a sequence of heel-off (HO), then, toe-off (TO) movements. The body moves forward further because of the push from the foot. The foot is swung in the air before the next initial contact. However, the walking gait may be disturbed due to stroke symptoms. For instance, foot drop causes an inappropriate foot position during the swing phase, which leads to the IC being on the toe instead of the heel. In the worst case, the patient may even suffer from an inability to swing the foot at all. Therefore, training for a normal walking gait must be done to recover the patient's walking gait.

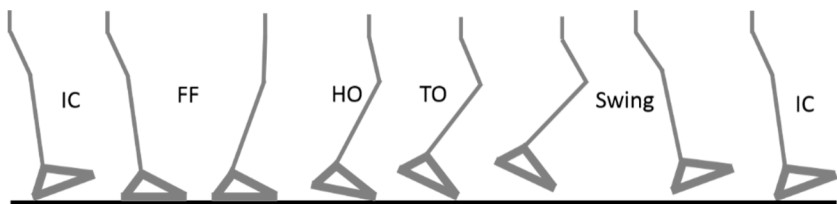

**Figure 1.** Walking gait classification. Initial Contact (IC), Foot Flat (FF), Heel-Off (HO), Toe-Off (TO).

Conventionally, walking gait training is performed manually with the assistance of a stroke therapist. Two to three therapists may be involved in helping the patient to maintain balance and to perform walking by moving the foot. Ancillary equipment, such as a cane, orthosis, walker, theraband, and so on, can also be used to support the patient during the training [9]. The equipment should fit (not be over-sized nor too tight) the user's anatomy to ensure the user's comfort, especially the fixing of the foot and shank to the footplate and the AFO strut [10,11]. Nowadays, the ancillary equipment is enhanced with robotic technologies, such as an orthosis and exoskeleton, to make it more sophisticated for rehabilitation sessions. Not only can the bio-physical data be monitored, but the training process can also be automated with the addition of robotic technology with an emphasis on assist-as-needed training [12]. Therefore, it is now possible to replace most of the work of therapists in conventional training through the use of robotics-enhanced ancillary equipment. The good news is that therapists can now focus more on monitoring and improving the quality of rehabilitation [4].

In the past decade, researchers have been able to develop many types of rehabilitation equipment that have been enhanced with robotic technology. One such piece of equipment is the Ankle-Foot Orthosis (AFO), which is usually used to hold in position an ankle that is weak due to spasticity or foot drop, and in some cases, to relieve pain in the foot of the patient [13,14]. Initially, a rigid AFO was developed to assist with dorsiflexion, thereby removing unnecessary restrictions on the plantar flexion direction. The material used was changed to thermal plastic material such polypropylene, which has the flexibility to bend. Less restriction on the plantar flexion was achieved without

decreasing the dorsiflexion assistance. However, the flexible rigid AFO was unable to replicate normal walking [15]. Then, the articulated AFO was developed with certain actuators installed on it, to control the mechanical properties of the AFO, thus producing a normal walking gait to help in training users. Each orthosis has its own specific gait control strategy within the framework, as shown in Figure 2. The input is calculated to obtain the output reference. Then, the output reference and output feedback are used to control the output, which comprises the desired mechanical properties.

Based on the estimation method for the controller output reference, the control strategies can be categorized into two groups, namely, subject-oriented (Figure 2a) and phase-oriented (Figure 2b) control types. The subject-oriented control strategy considers that the subjects will have different output references. For instance, patient A will have a different body weight from patient B, and thus, the output reference for the ankle stiffness of the AFO for each patient will not be the same [16]. On the other hand, the phase-oriented control strategy considers the gait to be divided into several phases. Then, the control output reference is treated differently according to the current ongoing phase selected by the selector. For example, the ankle stiffness is maximized to lock the foot in the swing phase, and is minimized to allow forward propulsion in the stance phase [17]. Both control strategies have their own pros and cons. The subject-oriented control strategy is more complex, but it involves only one calculation, because it must consider the entire walking gait. On the other hand, the phase-oriented control has more calculations due to the number of gait phases, but these are relatively simpler calculations. These must be considered because they will affect the type of equipment, sensing units, and actuators.

The mechanical properties of the AFO are the output references in the gait control, which is carried out either by a mechanical or electrical approach. In the mechanical approach, an actuator is used, which can be adjusted by a manufactured [18], modular [19], or screw mechanism [20]. The electrical approach uses an actuator that can be adjusted by an algorithm inside a microcontroller, computer, and so on. Several output references have been presented in the existing works such as bending stiffness, damping stiffness, assistive torque, and motion. An AFO with a damping stiffness control already offers advantages such as optimized weight, cost, and safety, because the actuator is a passive actuator [21]. Nevertheless, the damping stiffness control has not been optimized by using adaptive output references. The controlled damping stiffness has been adapted to different gait phases; however, it has not yet been adapted to different persons and environments. In addition, the adaptive output reference for knee damping stiffness has been conducted by using the Ground Reaction Force (GRF), but this has not been carried out yet for ankle damping stiffness.

This paper provides an overview of the control of the mechanical properties of existing AFOs for gait assistance in post-stroke patients. This paper is organized into a few categories, which are (1) AFOs covering the foot and shank, (2) modification of the AFO ankle joint with or without an actuator, and (3) potential applications for the rehabilitation of post-stroke patients. In addition, works on other limb-assistive devices such as the knee AFO, exoskeleton, and prosthesis, are also included to provide insights into gait control with that particular device. Several aspects that will affect the control of the mechanical properties of the AFO, such as the controller input, the number of gait phases, the controller output reference, and the controller performance evaluation, are discussed and compared. First, the discussion begins with a brief introduction about the types of AFO structures. Then, the controller input and the number of gait phases are discussed. The discussion continues with the estimation methods for the controller output reference, which is the main focus of this study. Based on the estimation method, the gait control strategies can be classified into subject-oriented estimations and phase-oriented estimations. Finally, suggestions for future works are addressed, one of which is the application of an adaptive controller output reference for maximizing the benefits of the AFO to users.

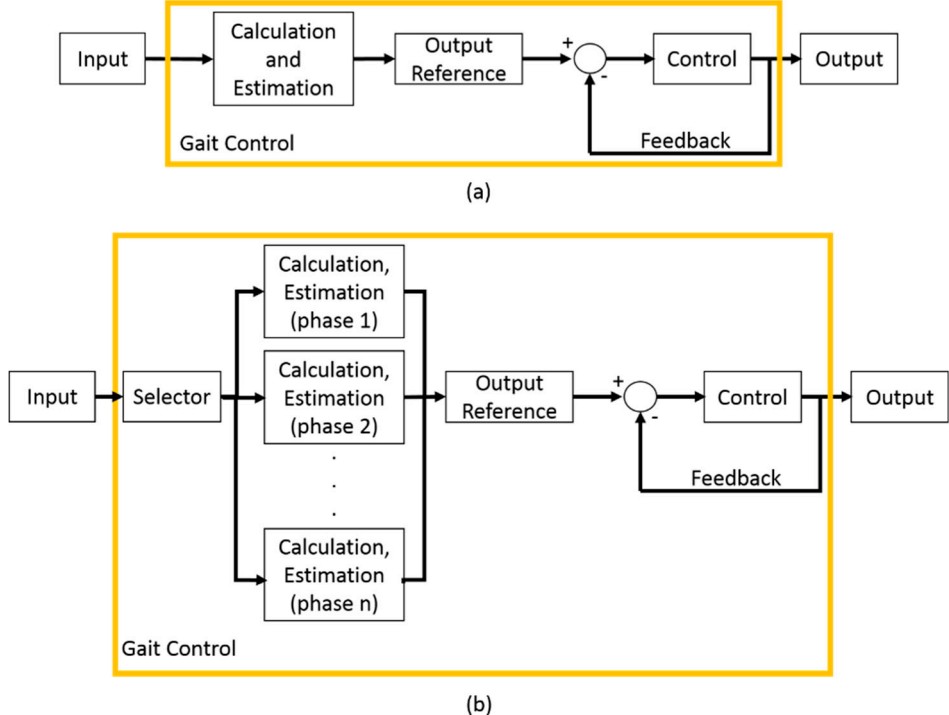

**Figure 2.** Gait control strategy framework: (**a**) Subject-oriented; (**b**) Phase-oriented.

## 2. Types of AFO Structures

Based on the structure of the joint, AFOs are classified as rigid AFOs, flexible rigid AFOs, and articulated AFOs, as shown in Figure 3. Initially, the AFO was made to assist the dorsal flexion during the swing phase by having a fixed rigid ankle joint. Thus, it was called a rigid AFO. The lack of dorsal flexion during the swing phase led to an inappropriate IC in the next stance phase, thereby giving rise to the risk of stumbling. Although dorsal flexion was successfully assisted, however, the forward ability during the stance phase decreased significantly [22]. Then, researchers improved on the AFO by giving it a controllable ankle joint, thereby giving rise to the flexible rigid AFO and articulated AFO. The flexible rigid AFO is made of a material that can bend, such as polypropylene [23]. Thus, the mechanical properties can be adjusted according to certain stiffness levels. Examples of the qualitative terms used to represent the stiffness levels include flexible, semi-rigid, and rigid [24]. Although, the flexible rigid AFO has no effect on the power output of the limb in the horizontal axis [25], it affects the power output in the vertical axis because of unnecessary restrictions on the plantar flexion, which is comparable to the rigid AFO [15,26]. Meanwhile, the articulated AFO is equipped with an actuator for controlling the mechanical properties according to the gait control strategy, and thus, it is more useful compared to the flexible rigid and rigid AFOs [27].

Several kinds of actuators such as Direct Current (DC) motors [28], pneumatics [29], magnetorheological (MR) devices (dampers [30] and brakes [17,31]), solenoids [32], and springs [20] control the gait, either by generating a movement (active AFO) or limiting the movement (passive AFO). By controlling the ankle joint, a healthy locomotion can be replicated, and the patient can be trained to walk normally [33]. An active AFO has more functions, such as the generation of movements [30] and the balancing of the body [34], than a passive AFO, in terms of its complex structure and algorithm. On the other hand, although the function of a passive AFO is limited, such as the control of the stiffness of the ankle joint only, it has a more compact structure and simpler algorithm [35]. The choice of the type of AFO depends on the patient's disability. Post-stroke patients with severe disabilities, for example, those who are unable to stand by themselves, may have to undergo advanced treatment such as neuromuscular electrical stimulation (NMES) [36], and utilize an active AFO [37]. However, for less

severe disabilities, such as foot drop [21], ankle arthritis [38], and the ankle giving way [39], the patient can benefit from the use of a passive AFO alone, instead of having to use an active AFO [21].

Not only actuators, but also sensors, such as a rotary encoder [17,21], accelerometer [40], foot switch [41], and electromyography (EMG) [17], are used together with AFOs for physical and biomedical data measurements. This information provides insights to therapists for improving stroke rehabilitation [42]. In terms of the gait control strategy, the information from the sensors can be utilized for the detection of the gait phase.

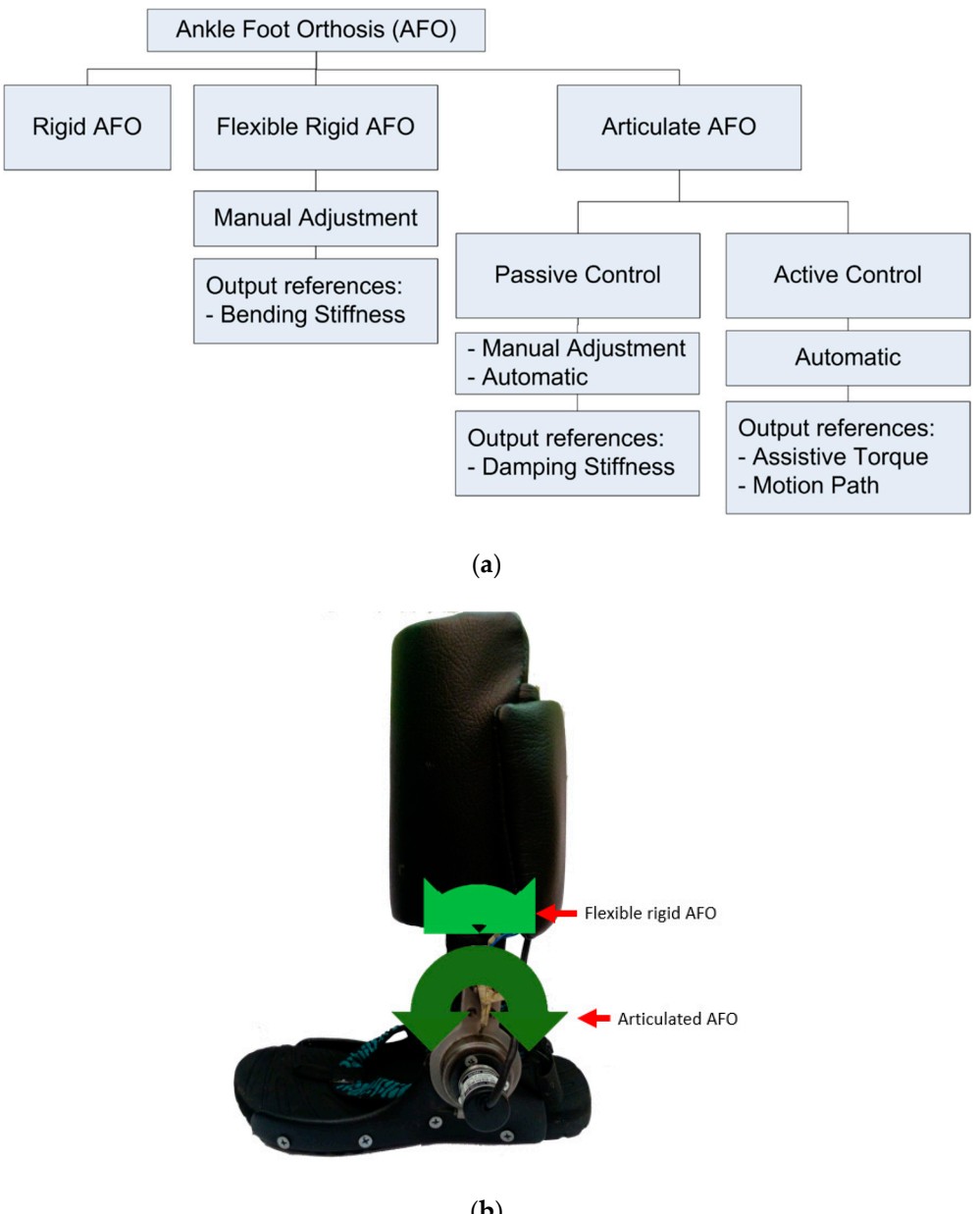

(a)

(b)

**Figure 3.** Hierarchical structure of Ankle Foot Orthosis (AFO) (**a**) and illustration of the AFO structure (**b**). Rigid, flexible rigid, and articulate AFO. The rigid AFO has no rotational movement; the flexible rigid AFO has a limited angle of rotation, and the articulated AFO has the largest angle of rotational movement of up to 360°.

### 3. Input and Gait Phases

The first aspects that are considered in the control of the mechanical properties of an AFO are the input and number of gait phases. The input for a control system is usually obtained from the users or their interaction with the environment. The reported inputs from the user include the EMG signal [43–50], joint angle [51], limb acceleration [21,40,51], bending moment [41,52], and the user's intention [30]. The joint angle can be obtained from the knee, hip, and ankle, depending on the purpose of the controller. The EMG is measured from the skin surface of the moving limb [53], while the limb acceleration is measured from the leg or foot translation acceleration. The bending moment is obtained from the AFO strut bending during usage. The user's intention is obtained by using push buttons, such as button 1 for walking mode 1, and so on. On the other hand, the ground reaction force (GRF) and the ground contact are kinds of user interactions with the environment that can be measured. The GRF and ground contact are measured by installing force sensors and foot switches, respectively, beneath the sole of the AFO.

The gait movement is classified into several phases to facilitate the control process. The input signal information guides the control system to correctly determine the current active gait phase. In other words, an inappropriate gait phase detection will result in misleading output references or mechanical properties of the AFO. For example, the timing control for storing and releasing the energy of a leaf spring to assist in the walking gait was shown in the work by Wilk et al. [32]. The leaf spring should store the energy during the stance forward propulsion, while the stored energy should be released when pushing off. As the timing depends on the gait phase, if the gait phase detection is not accurate, the timing will be disturbed. In other words, instead of assisting the gait, the AFO might become a burden instead.

In general, a gait may be classified into two phases, the stance phase (phase 1) and the swing phase (phase 2), as shown in Figure 4 [32,49]. In other existing works, the swing phase remains the same, but not the stance phase. The sequence for the contact between the foot and the ground can be further classified as IC to FF (phase 1, heel strike), FF to HO (phase 2, mid/forefoot strike), and HO to TO (phase 3, foot-off), thereby making it a 4-phased gait classification by the addition of a swing phase [41,54]. Another reported gait classification treats phases 2 and 3 in the 4-phased gait as a single phase, thus giving rise to a 3-phased gait classification in total: IC to FF (weight acceptance), FF to TO (stance termination), and a swing phase [30,32,55]. Not only is the gait phase classification addressed, but some reported control strategies include a walking mode classification such as sitting, standing up, and walking [30]. In another work, the incline walking and ascending–descending stairs are also considered in the control strategy [56]. The implementation of a walking mode has a wider application for gait control. However, it may need additional sensors for detection, thus increasing the complexity of both the structure and the algorithm.

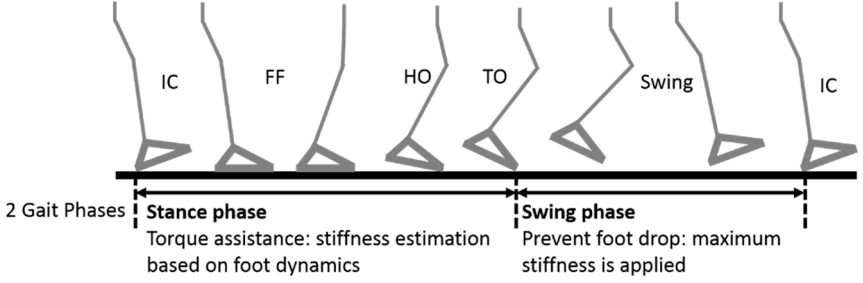

**Figure 4.** Reported gait phase classifications, namely, 2, 3, and 4-phased gaits.

The gait phase number is related to the input from the sensing unit on the AFO. If the input number of "x" is able to classify the gait into "n" number of phases, then the gait phase number is "n", and vice versa. As reported in a previous work by Gastrocnemius [17], the basic gait phases, such as the stance and swing phases, can be classified by using the active–inactive value of a single

EMG. A foot switch and force sensor installed on the bottom of the heel and toe to measure the foot contact can classify the gait into four phases [52]. The reading from a rotary encoder, combined with an accelerometer, can measure the threshold value for classifying the gait into three phases. The threshold indicates the beginning of each phase, as reported by Kikuchi et al. [40] and Svenson et al. [56].

A subject-oriented control strategy does not include the gait phases. In the case of a control strategy without the gait phases, the input is directly used to determine or to estimate the desired output reference. For example, pneumatic actuators were activated proportionally according to the corresponding magnitude of the EMG, as shown by Ferris et al. [47,53]. In another work, the EMG was used to estimate the desired motion by using a forward dynamics calculation [46].

## 4. Output Reference Estimation and Control

The AFO prescription can be unique for each application. In terms of the control strategy, the prescription means that the output reference is according to the capability of the actuator. The output of the control strategy is one parameter that should be controlled in order to improve the gait assistance. The parameters may include indirect variables or the mechanical properties of the AFO, including the bending stiffness [18], damping stiffness [56], and assistive torque [54], or direct variables related to the user kinematics, such as joint angle and velocity [21]. The user may not experience the benefits of using the AFO if there is a mismatch between the mechanical properties of the AFO and the user's kinematics [16]. Therefore, this output reference value must be estimated before the output is controlled. In this section, several existing output reference estimation methods are presented.

### 4.1. Bending Stiffness Control

A bending stiffness control was implemented on a flexible rigid AFO. The AFO was able to bend to a certain degree as such AFOs are made from a thermal plastic material such as polypropylene [18,57], and carbon fiber [19]. The works reported AFOs with different bending stiffness values such as 0.19 N·m [23] and 2.52 N·m [58]. Both bending stiffness values, which were not estimated and tested, showed that the AFO was not beneficial for all of the participating subjects. Therefore, the bending stiffness of an AFO should be controlled according to the needs of the user to optimize the benefits of using the prescribed AFO. The bending stiffness can be varied (i.e., 50%, 100%, and 130%) by changing the strut width, as shown in Figure 5.

Estimations of the bending stiffness can be made by using both a model approach [18] and a trial-and-error approach [59]. There are two steps of modelling for the model approach, namely, a user bending stiffness model and a strut width model. First, the user bending stiffness is accommodated with a musculoskeletal model. This model estimates the natural pseudo ankle stiffness of the user, which will then become the bending stiffness reference for the AFO. Then, the AFO is custom-manufactured by following the reference [60]. The bending stiffness depends on the material, and it is only known by measurements after the manufacturing process. The strut width model is used in this process to estimate a suitable strut width according to the bending stiffness reference. Schrank et al. [61], in his work, showed a strut width model called Virtual Functional Prototyping (VFP). Using this method, the Computer Aided Design (CAD) design of the AFO can be examined in a Finite Element Analysis (FEA) environment to obtain the bending stiffness reference. It should be noted that the real mechanical properties of the material must be used in the CAD software to avoid a misleading bending stiffness reference [62]. When the strut width has been obtained, the manufacturing process can be accurately performed by using an additive manufacturing technique such as selective laser sintering (SLS) [63].

For the trial-and-error approach, the strut width is set beforehand in the module [19,59,64], as shown in Figure 6. The strut module, which is made of carbon fiber, connects the calf and foot parts by a screw mechanism that can be easily and quickly changed, when necessary [19]. First, the bending stiffness of the strut module is identified, and then the parts have to be installed one-by-one, to determine which is the best one for the user. There will a change in the bending stiffness, not

only when a different user uses the AFO, but also when the user's condition improves or changes. For instance, if during the rehabilitation, a patient's condition improves, then the patient will need a different bending stiffness setting. Therefore, the bending stiffness reference needs to be adjusted from time to time, either quickly, by a modular approach [65], or accurately, by a manufacturing approach [63].

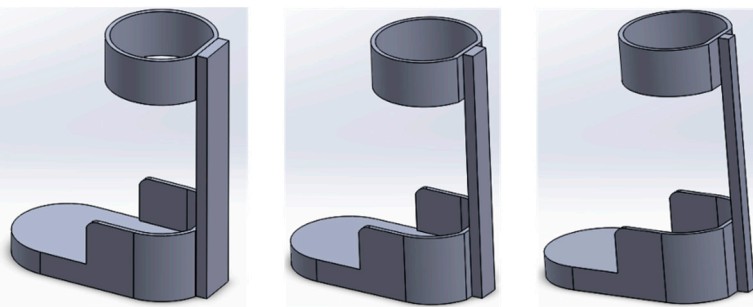

**Figure 5.** AFOs with different manufactured strut widths. From the left to right—stiffness of 130%, 100%, and 50%, respectively [61].

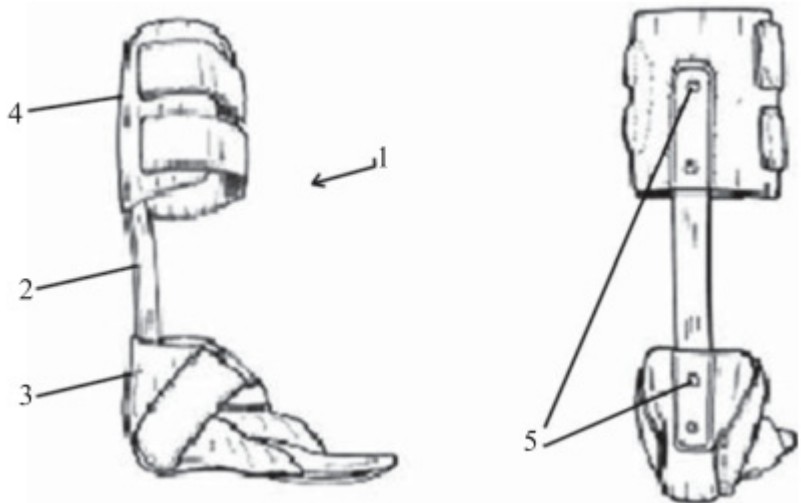

**Figure 6.** AFO with a strut module system: (1) Velcro strap; (2) strut module; (3) foot plate; (4) shank holder [19].

Note that a mechanical approach, such as through manufacturing and a strut module, instead of an electrical approach, is used to perform the bending stiffness control. The adjustment of the device is also not conducted in a real time application, but it is adjusted prior to the AFO usage. Therefore, it is inaccurate to assume that the bending stiffness control of an AFO is a control strategy. Nevertheless, the idea of bending stiffness control for an AFO has inspired the development of other control strategies.

*4.2. Damping Stiffness Control*

Damping stiffness control methods are classified in terms of the articulated type of AFO. The 360-degree range of motion makes it possible to have more gait control options, such as control in the dorsiflexion direction, plantar flexion direction, or dual direction control. Therefore, the users of AFOs with damping stiffness control will feel less hindered when walking by using the mentioned AFO [65]. The previous works that were reported presented both mechanical and electrical approaches for the damping stiffness control. In the mechanical approach, oil dampers and springs were used as actuators. In the electrical approach, motors, pneumatics, solenoids, and magnetorheological devices were used as actuators. In some works, springs were also presented in the electrical approach.

The mechanical approach to the control of damping stiffness is a subject-oriented control strategy utilizing an oil damper and spring. The oil damper has a hydraulic system with a spring inside to provide a resistance torque in one direction [66]. In the cases where more than one direction is needed, additional oil dampers need to be installed opposite each other [67], as shown in Figure 7a. The output resistance torque can be adjusted by a screw mechanism [68] and it may be classified into several levels [69]. The amount of stiffness in each level is measured beforehand by using a custom-made AFO stiffness measurement device [69]. The combination of the stiffness level on dorsiflexion and plantar flexion oil damper is configured according to the user, thus maximizing the effect of the stiffness of the AFO on the user's experience.

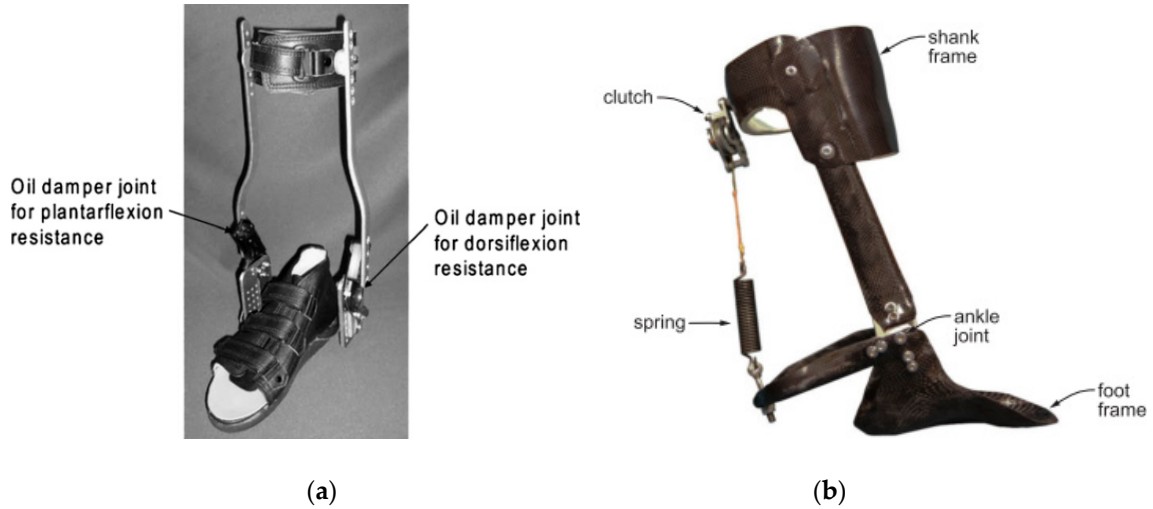

(**a**)            (**b**)

**Figure 7.** AFO with damping stiffness control. (**a**) Oil damper [67]; (**b**) spring [20].

It has been reported that a spring can also be used as an actuator to control the ankle joint stiffness, as depicted in Figure 7b. A spring has damping characteristics, and it is much lighter than an oil damper. It can store and release energy, and as such, it is able to assist the gait in both the dorsiflexion and plantar flexion directions [70]. Therefore, when a spring is utilized [20], the spring constant is the sole output reference that needs to be estimated and controlled accordingly. The spring constant may be controlled either by stretching it or even by changing the spring modulus [71]. Moreover, combinations of two or more springs can be used to add more variations of joint stiffness [72]. For instance, one spring was located at the front and one at the back of the foot, as shown in a study by Kobayashi [73]. Thus, it can be used to optimize the mechanical properties of an AFO for the individual gait treatment of post-stroke patients [74].

The estimation of the damping stiffness reference in real time was presented in the Knee AFO (KAFO) by Pete et al. [75]. The estimation is needed because the actual knee joint torque could be measured directly. In addition, an additional sensor for measuring the knee joint torque would have increased the weight of the AFO. Therefore, a model-based approach was chosen to determine the actual knee joint torque reference, which then became the damping stiffness reference. The model was based on the ground reaction force (GRF). It is much simpler to measure the GRF by using a force sensor installed at the palm of the AFO. This sensor does not add significant weight to the AFO. Variations in the GRF measurements can be done, such as all foot GRFs, toe GRF only, or heel GRF only. The force sensor location must be considered carefully, so that the GRF can always be measured without the need for the entire foot to be in contact with the ground all the time.

The damping stiffness control can also be applied differently for each gait phase by emphasizing the stance phase assistance [76]. The work by Wilk et al. on the ADJUST demonstrated an AFO with a spring actuator (leaf spring), plus an additional locking feature by using a solenoid, depending on the gait phase, as shown in Figure 8 [32]. Another work showed a more advanced gait phase-dependent

damping stiffness control using a hydraulic damper [77]. The KAFO basically has two neural modular controllers: gait phase tracking, and a gait prediction and selection controller. Initially, an example of a gait pattern is planted in the controller. The gait phase tracking maps the detected gait phase, according to the GRF and joint angle (hip and knee), to the desired damping stiffness, based on this gait example. The second modular controller then predicts the current gait phase by learning from the current information and the gait example. Thus, by having this prediction step, the controller is able to adapt to changes in the gait. Finally, after the prediction, the gait phase with the best fit will be chosen to implement the desired damping stiffness.

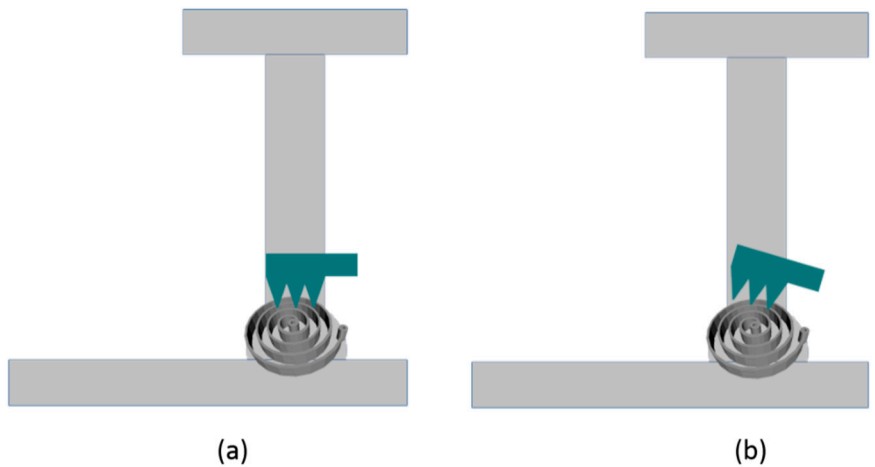

**Figure 8.** Adjust by Wilk. (**a**) Locked position; (**b**) release position [32].

The stiffness level can be controlled by the means of a magnetorheological (MR) device [35]. An MR device can produce a proportional degree of stiffness through an applied current. The reported MR devices are the damper [56], which works in a translational movement, and the brake [17,31], which works in a rotational movement. By installing such a device to an AFO, the damping stiffness can be controlled as desired.

The output references are different for different gait phases, as shown in the work on an Intelligent AFO (I-AFO) by Kikuchi et al. [21,40,41,52,78]. Figure 9a shows the damping stiffness reference for an MR brake, which was inspired by the I-AFO, for the purpose of preventing foot drop. Refer to Figure 4 for the details on the phases. In phase 1, "damp", the damping starts at a high level, but it decreases along the phase for damping the foot movement from IC to FF. In phase 2. "free", no damping stiffness is applied, to allow for forward propulsion from FF to HO. A "free" damping stiffness is also applied in the descending stairs mode when the foot fell [56]. In phases 3 and 4, "lock", the damping stiffness is static in the middle-high value, which is enough to lock the foot position and prevent foot drop. As for the exact value of the torque, it can be set beforehand according to the subject's needs.

Another work on the AFO with an MR brake, known as the Passive Controlled AFO (PICAFO), was conducted by Adiputra [48–50]. The purpose of the control was the same as for the I-AFO, which was to prevent foot drop on post-stroke patients. However, the damping stiffness reference for the MR brake was not a fixed value, but a fuzzy one [48]. The ankle position and EMG signal became the input for a fuzzy-based controller (FC) to estimate the voltage output for generating the damping stiffness of the MR brake, as shown in Figure 9b. The foot drop was expected to be prevented when using the PICAFO, which can be observed from the data on the ankle position. Thus, the FC was manually tuned by comparing the resultant ankle position with the desired ankle position [48]. For instance, membership functions included modifications for improving the maximum torque achievement [17] and modifications to the fuzzy rules for improving the accuracy of the torque to the gait phase [49].

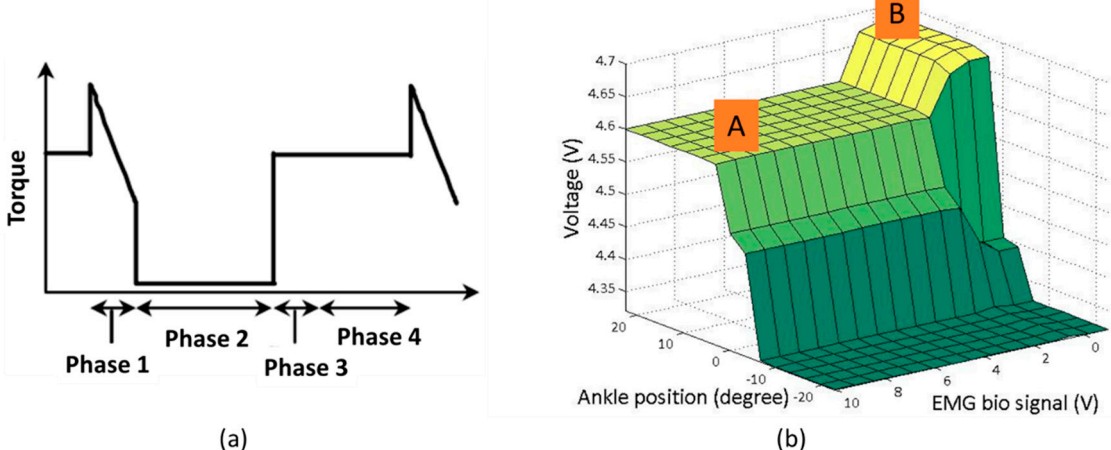

**Figure 9.** Torque references. (**a**) Fixed output reference [41]; (**b**) Fuzzy output reference [17].

### 4.3. Assistive Torque

An assistive torque helps the lower limb to move, balances the body, and thus performs walking. The actuators used for assisting the torque include a motor [28], a pneumatic system [29], and a series-elastic actuator [54]. The spring is basically a mechanical actuator that is limited to storing and releasing the potential spring energy. By combining the spring with a motor, it can perform the torque assistance in a combination called Series Elastic Actuator (SEA) [51].

The WAKE-up, an exoskeleton by Patane et al. [54], is equipped with a rotary SEA to provide the torque with assistance during a determined gait phase. The gait phase detection is done by using a foot switches sensor in a 3-phase gait. The assistive torque is controlled by using a Proportional Integral Derivative (PID) controller on each phase with feedback from a rotary encoder and an inertial measurement unit (IMU) on the rotary SEA. The WAKE-up provides torque assistance to the knee and the ankle, both of which are controlled in a modular configuration. With reference to Figure 4, the maximum torque on the knee module is exerted for flexion during the TO, and is exerted for extension during the remaining gait phase. As for the ankle module, it has a maximum torque for flexion during IC and TO (phases 1 and 3), and a maximum torque for extension during the mid/forefoot strike (phase 2). The modularity gives a degree of flexibility to the WAKE-up. For example, to control the ankle only like the AFO, the knee module can be removed [54].

A pneumatic is a type of actuator that works on the same principle as the muscle; therefore, it is usually used as an artificial muscle. The assistive torque from a pneumatic was presented in the work by Ferris et al. [47,53]. An EMG-force model was used to estimate the pneumatic force proportionally. Two muscle signals contributed to the model: The soleus EMG activating the plantar flexion pneumatic, and the tibialis anterior EMG activating the dorsiflexor pneumatic, as shown in Figure 10. This model is not limited to a walking scenario. If there is muscle activity, even outside a walking scenario, the controller will receive the pneumatic force reference estimated by the model. Then, the pneumatic will exert a controlled force accordingly. Different persons may have different EMG values. Therefore, if the force exerted by the pneumatic is not enough, then it can be easily tuned by adjusting the model gain. Not only will this help the patient to walk on their own will, but this control strategy of the EMG-force model also helps therapists and researchers to understand the improvement of the patient during walking [47]. Increasing the number of pneumatics, for instance, to four, can increase the accuracy of the trajectory control, but with drawbacks in the complexity of the structure [79].

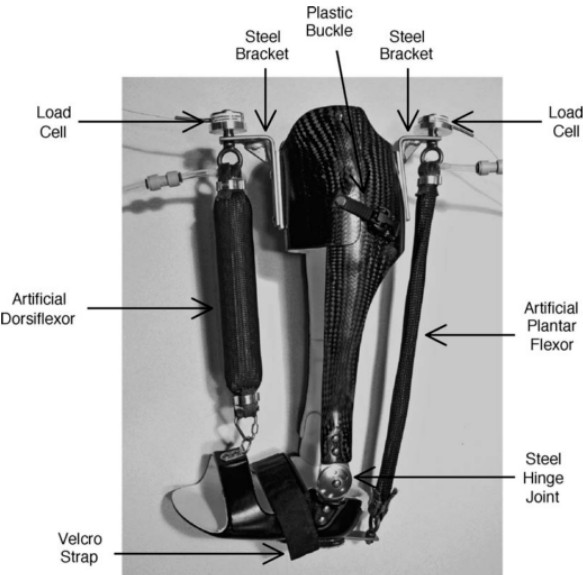

**Figure 10.** AFO with an artificial pneumatic muscle [48]. The front pneumatic assists the dorsiflexion, and the rear pneumatic assists the plantar flexion.

### 4.4. Motion Path Control

In the previous control methods, the output references are the mechanical properties of the AFO. The gait control can be performed well, but there is the risk of incorrect motion if no feedback comes from the motion. The motion control in an AFO relates to the position and velocity control of the ankle joint, as shown in the portable powered AFO (PPAFO) by Shorter [55]. There are three gait phases presented in the control strategy of the PPAFO, namely, phase 1 (IC to FF), phase 2 (FF to TO), and phase 3 (TO to IC), with different ankle position references. It is difficult to control the position by using an actuator when the output cannot be controlled. For instance, the torque exerted by a solenoid valve cannot be controlled. Therefore, a proportional valve must be used to rotate the PPAFO ankle joint because of the ability to vary the exerted torque. A torque–ankle position model was presented in the study, and the amount that was gained could be adjusted depending on the subject, the walking mode, and so on.

The position control not only requires the accurate generation of movement, but also the generation of the appropriate braking torque for stopping the movement. The movement may be generated by a motor. However, it would have been inappropriate for the braking torque to also be done by the motor. Therefore, an MR brake was suggested in the work by Chen et al. to provide the braking torque [30]. The combination of these actuators was then called as Magnetorheological Series Elastic Actuator (MRSEA). The energy cost for activating the actuator can be saved by using an MR brake. However, in this work, the contribution of the MR brake was not fully optimized.

In another work, a velocity control using an MR brake was presented in the third development of the I-AFO by Kikuchi et al. [78]. First, the gait was changed from four phases to three phases, based on the joint angle and the accelerometer threshold. During the heel strike, from IC to FF, there was the probability of a foot slap occurring for a post-stroke patient. A foot slap occurs when the foot, at the IC, performs the plantar flexion too fast. Because of this, an appropriate ankle velocity must be controlled during this phase. A PID controller was implemented for controlling the velocity. The ankle velocity reference is different from one person to another. Thus, not only the threshold value for the gait phase selector, but the ankle velocity reference also has to be set according to the subject [21].

The motion control usually needs to follow an example, as shown in the work by Guerrero et al., where 20 healthy subjects participated to provide a gait example [80]. In the case of the absence of an example, it is necessary to predict the intended motion. The prediction of the intended motion was presented by Fleischer et al. [46]. Unlike Ferris, the EMG-force model was used in the higher control

layer, together with a biomechanical model instead of controlling the actuator directly, as shown in Figure 11. The joint angle information was processed by using an inverse dynamic calculation to identify the current active torque. The EMG-force model was used to estimate the upcoming muscle force. The EMG-force model parameter was calibrated in real time based on the currently active torque and the measured EMG. A forward dynamic calculation was performed by using force estimation, active torque, and joint angle, to determine the desired motion or motion reference. The motion controller for the actuator translated the desired motion into a servo motor rotation to allow the user to walk.

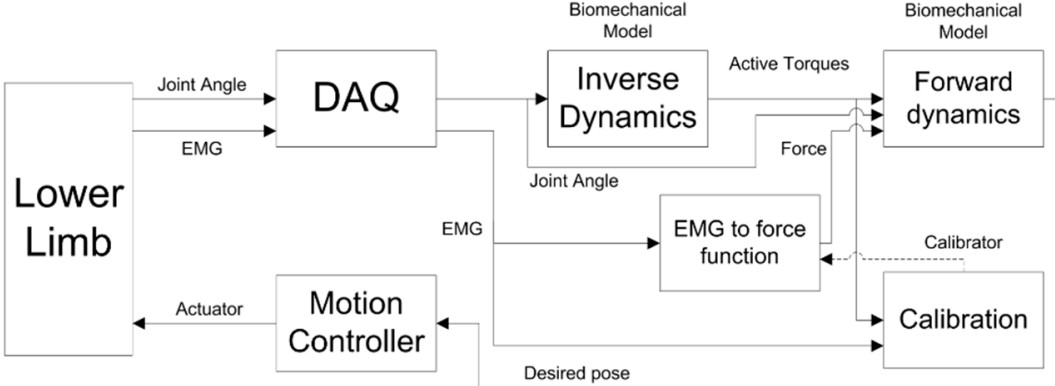

**Figure 11.** EMG-force model for the desired position estimation [46]. A biomechanical model was used for the inverse and forward dynamics. Calibration was done on the EMG to force the function.

Another important factor when performing motion control is the generation of a non-linear pattern of the trajectories, which is usually done by a central pattern generator (CPG) [81]. The CPG is a non-linear oscillator that works at a certain pattern frequency for different walking situations. The work by Nachstedt et al. showed a frequency adaptation for the CPG [82]. By applying the pattern frequency adaptation, the system could detect and tolerate an external pattern frequency from the disturbances over a considerable length of time [83]. Theoretical information about the gait is necessary for initializing the CPG, such as when the walking is periodic. The initialization of the CPG is not necessary in a system with EMG, because the EMG activation pattern is already being controlled by the brain without any additional information from the outside [84].

In cases of difficulty in using the EMG, such as inaccurate and inconsistent measurements [85], a virtual EMG can be the alternative in the AFO, as shown, and it can be controlled by using the Neuromuscular Control (NMC) method. The NMC is a control strategy that compromises several control layers, such as the body mechanics (BM) layer, muscle actuation (MA) layer, neural control (NC) layer, and higher control (HC) layer [86], as depicted in Figure 12. It has been implemented in several supportive devices such as the gait trainer [87], Active Pelvis Orthosis (APO) [85], and prostheses [86]. In the NMC, the virtual EMG is modelled to exert a virtual muscle force in the MA layer according to the Hill-type muscle model [88]. Then, the timing of its activation is controlled by a stimulus from the NC layer. The stimulus activates the hip and knee muscles during the swing phase, and the ankle and knee muscle during the stance phase [89]. The stimulus will be in a non-linear pattern, and is generated as a muscle reflex to balance the body. As such, the NMC is applicable for a wide range of walking modes (ground level, inclination, ascending stairs, descending stairs, running, etc.) [90], with adaptations to environmental disturbances [91].

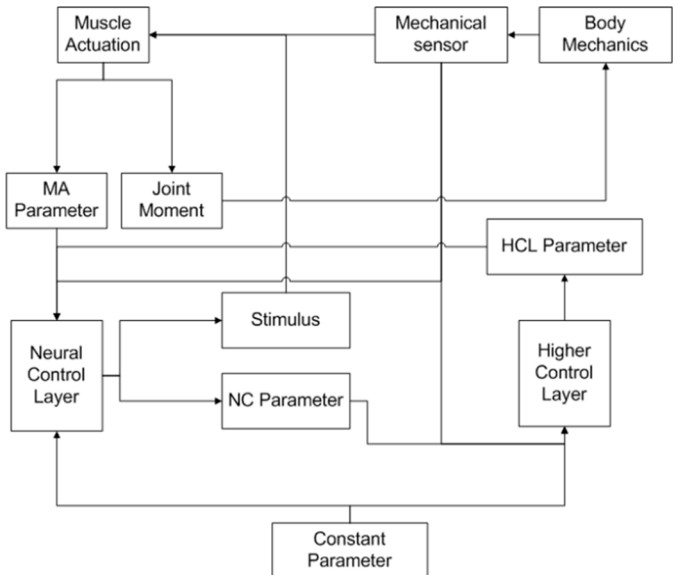

**Figure 12.** Neuromuscular control strategy framework [88]. There are four main parts: muscle actuation, body mechanics, neural control and higher control layers.

## 5. Discussion

Table 1 shows several reported works on the AFO. There are three types of AFO structures, namely, the rigid, flexible rigid, and articulated AFO. The articulated AFO is reported to have at least one actuator installed for controlling the joint, such as an oil damper, spring, solenoid, SEA, and MR device. There are two types of AFO control systems/strategies: subject-oriented and phase-oriented control strategies. The phase-oriented strategy divides a walking cycle into two, three, or four gait phases, using information from the sensing unit, such as the joint angle, EMG signal, limb acceleration, bending moment, and user intention. Then, the output reference on each phase is determined. As for the subject-oriented control, the input is used directly to estimate the output reference. This section discusses several interesting points that were considered when the reported AFO gait control strategies were being developed, such as the input and gait phase, output reference, and gait control performance evaluation.

**Table 1.** The comparison of the development of AFO over the past decade.

| Name | Control Orientation | References | Input | Gait Phases | Output Reference | Estimation Method | Control | Actuator | Performance |
|---|---|---|---|---|---|---|---|---|---|
| Rigid AFO | | 16, 22, 23, 38, 42 | - | - | Bending stiffness | - | Manufacturer | Thermoplastic | - Plantarflexion is restricted, resulting in less power generated during push-off<br>- Dorsiflexion is supported, resulting in the success of toe clearance. |
| Flexible Rigid AFO | | 18, 25, 26, 57, 58, 61, 63 | Motion | - | Bending stiffness | Model | Manufacturer | Thermoplastic | - Plantarflexion is less restricted, resulting in moderate power generated during push-off<br>- Dorsiflexion is supported, resulted in the success of toe clearance.<br>- Accurate prescription due to custom manufacturing. |
| | | 19, 59, 60, 64 | Motion | - | Bending stiffness | - | Modular | Carbon fiber | - Plantarflexion is less restricted, resulting in moderate power generated during push-off<br>- Dorsiflexion is supported, resulting in the success of toe clearance.<br>- Faster adaptation to disturbance because of modularity. |
| | Subject-Oriented | 62 | | | Bending Stiffness | Model | Manufacturer | Acrylonitrile Butadine Styrene (ABS) Polymer | - Plantarflexion is less restricted, resulting in moderate power generated during push-off<br>- Dorsiflexion is supported, resulting in the success of toe clearance.<br>- Faster manufacturing from a 3D printer. |
| Articulated AFO | | 28, 34 | GRF, Motion | - | Motion | Direct | PID | DC motor | - Plantarflexion and dorsiflexion are successfully supported.<br>- Position-tracking accuracy is good and can be neglected.<br>- Muscle activity decreased by about 8% to 29%<br>- Device energy cost is unidentified.<br>- Long-term and short-term effects are unidentified. |
| | | 80 | GRF, Motion | - | Motion | Direct | Lyapunov Control | DC motor | - Plantarflexion and dorsiflexion are successfully supported.<br>- Position tracking accuracy is good, with a position error of 5.42°<br>- Muscle activity is unidentified<br>- Device energy cost is unidentified.<br>- Long-term and short-term effect had not been studied |
| | | 29, 47, 53 | EMG | - | Assistive Torque | Direct | PID | Pneumatic | - Plantarflexion and dorsiflexion are successfully supported.<br>- Torque tracking accuracy is unidentified.<br>- Muscle activity decreased by about 50% to 60%<br>- Device energy cost is 40 J to 50 J.<br>- Long-term and short-term effects had not been studied |

**Table 1.** *Cont.*

| Name | Control Orientation | References | Input | Gait Phases | Output Reference | Estimation Method | Control | Actuator | Performance |
|------|---------------------|------------|-------|-------------|------------------|-------------------|---------|----------|-------------|
| Articulated AFO | Subject-Oriented | 20, 71, 72, 70, 73, 74 | Motion | - | Damping stiffness | - | Screw mechanism | Spring | - Plantarflexion and dorsiflexion are successfully supported.<br>- Muscle activity decreased by about 8%.<br>- No device energy cost because a non-power-consuming actuator is used.<br>- Long-term and short-term effects had not been studied |
| | | 65, 66, 67, 68 | - | - | Damping Stiffness | - | Screw Mechanism | Oil damper | - Plantarflexion and dorsiflexion are successfully supported.<br>- Muscle activity is unidentified.<br>- No device energy cost because a non-power-consuming actuator is used.<br>- Short-term effect shows positive effects for the user. |
| | | 79 | Motion | - | Motion | - | PID | Pneumatic | - Plantarflexion and dorsiflexion support is unidentified.<br>- Good position tracking accuracy with a Root-mean-square Deviation (RMSD) of 0.0065 to 0.0714<br>- Muscle activity is unidentified.<br>- Device energy cost is unidentified.<br>- Long-term and short-term effects are unidentified. |
| | Phase-Oriented | 17, 48, 49 | EMG, Motion | 2 | Damping stiffness | Fuzzy Logic | - | MR brake | - Plantarflexion and dorsiflexion are successfully supported.<br>- Accuracy is unidentified.<br>- Muscle activity is unidentified.<br>- Device energy cost is unidentified.<br>- Long-term and short-term effects have not been studied. |
| | | 31, 56 | Foot Switch, Motion | 3 | Damping stiffness | Fix look-up table | - | MR damper | - Plantarflexion and dorsiflexion are successfully supported.<br>- Accuracy is unidentified.<br>- Muscle activity is unidentified<br>- Device energy cost is unidentified.<br>- Long-term and short-term effects have not been studied. |
| | | 32 | Motion | 3 | Damping stiffness | Model | On/off | Solenoid | - Plantarflexion and dorsiflexion are supported, but uncomfortable.<br>- No accuracy is necessary.<br>- Muscle activity is unidentified.<br>- Device energy cost is about 184 watts.<br>- Long-term and short-term effects have not been studied. |

**Table 1.** *Cont.*

| Name | Control Orientation | References | Input | Gait Phases | Output Reference | Estimation Method | Control | Actuator | Performance |
|---|---|---|---|---|---|---|---|---|---|
| Articulated AFO | Phase-Oriented | 21 | Motion | 3 | Motion | Model | PID | MR brake | - Plantarflexion and dorsiflexion are successfully supported.<br>- Adaptive torque tracking reference.<br>- Muscle activity decreased by about 60%<br>- Device energy cost is unidentified.<br>- Long-term and short-term effects have not been studied |
| | | 40, 78 | Motion | 3 | Motion | Direct | PID | MR brake | - Plantarflexion and dorsiflexion is successfully supported with an emphasis on foot slap.<br>- Torque tracking is good, but the reference is fixed and not suitable for all subjects.<br>- Muscle activity is unidentified.<br>- Device energy cost is unidentified.<br>- Long-term and short-term effects have not been studied. |
| | | 41, 52 | Foot Switch, Motion | 4 | Damping stiffness | Fix look up table | - | MR brake | - Plantarflexion and dorsiflexion are successfully supported.<br>- No position reference is determined.<br>- Muscle activity is unidentified.<br>- Device energy cost is unidentified.<br>- Long-term and short-term effects have not been studied. |
| | | 55 | Motion | 3 | Motion | Direct | PID | Solenoid | - Plantarflexion and dorsiflexion effects are untested.<br>- Good position tracking, but slow response.<br>- Muscle activity is unidentified.<br>- Device Energy cost is unidentified.<br>- Long-term and short-term effects have not been studied. |
| KAFO | Subject-Oriented | 46 | EMG, Motion | - | Motion | Model | PID | Motor | - Plantarflexion and dorsiflexion are not successfully supported.<br>- Inaccurate position tracking and slow response.<br>- Muscle activity is unidentified.<br>- Device energy cost is unidentified.<br>- Long-term and short-term effects have not been studied. |
| KAFO | | 75 | GRF | - | Damping stiffness | Model | PID | SEA | - Sit-to-stand movement is successfully supported.<br>- Good torque tracking with a response time of less than 50 ms.<br>- Muscle activity decreased by about 23%<br>- Device energy cost is un-identified.<br>- Long-term and short-term effects have not been studied. |

**Table 1.** *Cont.*

| Name | Control Orientation | References | Input | Gait Phases | Output Reference | Estimation Method | Control | Actuator | Performance |
|---|---|---|---|---|---|---|---|---|---|
| KAFO | | 51, 54, 76 | GRF, Motion | 3 | Assistive torque | Direct | PID | SEA | - Plantarflexion and dorsiflexion are successfully supported.<br>- Position tracking is not good as it is not the same as the reference.<br>- Muscle activity is unidentified.<br>- Device energy cost is unidentified.<br>- Long-term and short-term effects have not been studied. |
| KAFO | Phase-Oriented | 77 | GRF, motion | 2 | Damping stiffness | Model | Neural Network | Hydraulic Damper | - Plantarflexion and dorsiflexion are successfully supported.<br>- Adaptive accurate torque tracking with a self-learning algorithm.<br>- Muscle activity is unidentified.<br>- Device energy cost is unidentified.<br>- Long-term and short-term effects have not been studied |
| Exoskeleton | | 30 | GRF, Motion | 3 | Motion | Direct | PID | MRSEA | - Plantarflexion and dorsiflexion are successfully supported.<br>- Good position tracking.<br>- Muscle activity is unidentified.<br>- Device energy cost of the motor is decreased by 52.8% to 95.5% of 411 J. However, the whole MRSEA energy output has not been investigated.<br>- Long-term and short-term effects have not been studied |
| Exoskeleton | | 84, 87 | GRF, Motion | - | Motion | Model | - | SEA | - Can be used for a person who is unable to walk at all.<br>- Good position tracking and torque tracking.<br>- Muscle activity is substituted by the device.<br>- Device energy cost is unidentified.<br>- Long-term and short-term effects have not been studied. |
| Active Pelvis Orthosis | Subject Oriented | 85 | GRF, Motion | - | Motion | Model | - | Motor | - Supports walking by moving the thigh.<br>- Good adaptive position tracking.<br>- Muscle activity is substituted by the device.<br>- Device energy cost is unidentified.<br>- Long-term and short-term effects have not been studied. |
| Prosthesis | | 86 | GRF, Motion | - | Motion | Model | PID | SEA | - Can be used for people who are unable to walk at all.<br>- Torque reference is automatically produced by the NMC.<br>- Muscle activity is unidentified.<br>- Device energy cost is unidentified.<br>- Long-term and short-term effects have not been studied. |

### 5.1. Input Considerations for Gait Phase Classification

Although the articulated AFO joint can rotate 360°, the actual ankle joint does not rotate much during walking, either in the plantar flexion or dorsiflexion directions. Different walking modes require different ranges of motion. For instance, walking uphill requires a greater dorsiflexion angle, while walking downhill requires more plantar flexion. A customized limitation of dorsiflexion and plantar flexion can be achieved by controlling the damping stiffness. If the damping stiffness is inappropriate, then the assistance may instead disturb the walking mode. For instance, too much damping stiffness on the dorsiflexion will disturb the pulling up of the foot when ascending stairs, and too much damping stiffness on the plantar flexion may prevent the toe from touching the stair when descending. The timing for the activation of a certain controller is related to the gait phase, making the classification of the gait phases an important task for the gait control of the AFO [56].

There has been no clear guidance reported on the choice of the best gait phase classification. The number of gait phases does not make one control strategy superior to another. However, the sensor configuration is considered when choosing the number of gait phases, because of the weight of the sensor. The more sensors that are installed on the AFO, the more information can be obtained; however, in some cases, the information may be not necessary for gait control. Instead, the sensors only add to the bulkiness of the AFO, thus increasing the weight. In other words, it is preferable to have fewer sensors. Hence, the bending moment and force sensor were replaced by foot switches to decrease the weight of the AFO [41]. The durability of the sensor was also taken into consideration. For example, the foot switches were changed to an accelerometer, because the foot switches were stomped on, while the accelerometer was not. This change lowered the number of gait phases from four to three [40].

The input from the EMG is very informative. Besides the classification of the gait phase [49], it can also be used to determine the output reference during walking without considering any gait phase. The EMG can be modelled to estimate the muscle force; thus, it is suitable for muscle-like actuators [43–45] such as a pneumatic [46,47,53]. The muscle contraction, which can be static or dynamic, can be modelled and analyzed as a non-linear polynomial function [44,45] or as parallel cascade identification (PCI) [43]. As for the gait control, it would be more suitable to implement a dynamic EMG model, because walking is a dynamic activity. The advantage of the EMG model is that the past input is not required, and thus, it does not need initialization. Also, it can be used for clinical diagnoses.

However, the control strategy used for the EMG faces a challenge in the sensing unit, such as for measurement accuracy and consistency [85]. Complex signal processing is also needed to use EMG features for gait phase classification [92]. The virtual EMG can be used as an alternative to the surface EMG for the application of gait control, because it is not measured, but modelled instead. However, the virtual EMG cannot be activated on its own. Therefore, when using the virtual EMG for gait control, the detection of the gait phase is important for determining the timing of the EMG activation [89].

### 5.2. Output Reference: Fixed versus Adaptive

The success of a certain control system/strategy for an AFO is not solely determined by the method used in the control system. The most important part is the method that is used to determine the output reference. Previous works have shown the output reference estimations for the bending stiffness, damping stiffness, and assistive torque. There are two kinds of output references; namely, fixed and adaptive output references. In the fixed one, the output reference is not changed in real-time applications, even if there is a disturbance. Meanwhile, in an adaptive output reference, the output reference could change in real time according to the current situation, and thus, it is more reliable against disturbance. The kinds of disturbances that may force a change in the output reference include the user/subject/patient, the walking mode, walking terrain, walking speed, and so on.

The reported control of the mechanical properties of an AFO, such as bending stiffness, damping stiffness, and assistive torque, showed both fixed and adaptive output references. It could be clearly observed that the fixed output reference worked with a flexible rigid AFO [18,64] and an AFO with only

mechanical actuators, such as an oil damper [69] and a spring [32,73]. The adaptive output reference could be observed from those works that utilized electrical actuators such as an MR device [41,49,56], SEA [54,75], and a pneumatic system [53]. The adaptive output reference was not adapted to all types of disturbances, but some of them, such as the gait phase [41] and GRF were adaptive output references [75]. The previous works also showed that when the EMG model was used in the control system, the output reference was able to adapt to more types of environmental disturbances.

Controlling the mechanical properties of the AFO does not guarantee that the desired motion will be achieved, but that it may be achieved by using motion control. Controlling the motion enables the actuators to be controlled; thus, the tracking of the position [30] or velocity [21] can be done. However, it would be better if the output reference is an adaptive one, especially in the stance phase [93]. Because there is only one reference, it is assumed that the position of the ankle joint is the same the entire time. When there is a disturbance, such as when the walking mode changes from ground level to an incline, the position reference should also be different. Therefore, it is important that the output reference, such as the position, be adapted to the disturbance in real time [54]. Several previous works have reported on adaptive motion controls, such as a velocity control [21], EMG-driven control [46], and NMC [86]. The reported velocity control was only implemented during the IC to FF, with the velocity reference being obtained from the processing of the information from the previous step. Therefore, initialization was needed for velocity control, such as a pre-determined velocity reference for the first step. Initialization was also needed for motion control by using the CPG [84]. Unlike velocity control, a motion control that uses the EMG (i.e., EMG-driven and NMC) can control the motion during the entire walking activity without any past information.

The adaptive output reference is without doubt superior to a fixed output reference in terms of disturbance tolerance. In addition, the adaptive output reference also reduces the work of the therapist or researcher in setting the output reference, whenever necessary. However, in terms of hardware management, a fixed output reference may have the upper hand. For example, a fixed output reference using an oil damper and spring has a lower electrical energy cost compared to an adaptive motion control by using a DC motor. Also, the adaptive output reference may require a more sophisticated sensing unit because additional information is demanded for the estimation of the output reference. A quick example is the force sensor versus the foot switches. Both sensors can give information about foot contact. However, a force sensor offers additional information such as the GRF, which can be used to estimate the desired joint torque [75,77]. When using a GRF sensor, the position of the sensor should be especially considered for obtaining the reading from the heel contact force [12]. Thus, it can be concluded that certain factors should be considered when choosing between an adaptive and fixed output reference for an AFO.

*5.3. Controller Performance Evaluation*

The controller performance evaluation of the AFO gait control is also something to be considered when developing the AFO. However, previous studies have reached a weak conclusion on the performance evaluation, due to a lack of high-quality research [94]. If a performance evaluation cannot be conducted, then the performance cannot be measured and compared. The performance of the controller is mainly conducted by rating the tracking accuracy of the output reference. However, the purpose of the AFO should not be forgotten, which is to perform normal walking for the user/patient/subject. Therefore, it is also important to consider the user kinematics, walking energy cost, device energy cost, and the long-term as well as short-term effects of the AFO on the user when assessing the gait control performance of the AFO.

**The tracking accuracy** shows the success rate of the AFO controller output in following the output reference. Reported work on the control of the mechanical properties of the AFO showed that accurate output reference tracking was obtained by using well-established methods such as the use of a PID controller or FC. However, for motion control, the tracking accuracy depends more on the actuator type. The tracking accuracy is higher if an actuator that can vary its output is used. The work

by Shorter shows the use of a solenoid valve for position control [55]. To improve the position control, the solenoid valve was replaced by a proportional valve, whereby the output could be varied.

**The user kinematics** shows the mechanical properties of the lower limb when performing walking, including the joint angle, joint angular velocity, and GRF. It will be easier to achieve the desired user kinematics by using the motion control. However, when applying the control of the mechanical properties of the AFO (bending stiffness, damping stiffness, and assistive torque), the user kinematics are indirectly controlled. Therefore, the optimization of the performance of the AFO with motion control should be easily done, rather than for an AFO, with control of the mechanical properties. However, the therapist or the researcher has to ensure that the output reference for the motion control is appropriate for the desired user kinematics.

**The walking energy cost** may increase because of inappropriate ankle movements; thus, gait control is performed. The energy cost can be determined based on the oxygen that is used by the user during walking. The reported AFO with electrical actuators mainly focused on the user kinematics by reproducing the desired walking motion in the AFO users. The walking energy cost was rarely mentioned, but it was highlighted in work on an AFO with a spring [20], and the simulation of the KAFO with a spring [95], which showed that healthy subjects who walked by using the AFO with a spring had their walking energy cost reduced. Another way to observe the energy cost is through the EMG activity during walking [28]. Work on the I-AFO showed that the MR brake was doing work that was equal to the work done by the rectus femoris muscle during walking [21]. Thus, during walking with the assistance of an MR brake, the rectus femoris activation was reduced, indicating that the energy cost exerted by the body decreased.

**The device energy cost** shows the energy used by the AFO during gait assistance. This was rarely discussed when assessing controller performance. For instance, the AFO spring already reduces the walking energy cost with just the mechanical actuators. This means that no electrical energy is needed by the AFO with an additional reduction in walking energy. On the other hand, the AFO with an electrical actuator needs an external energy source for the actuator itself. Therefore, the device energy cost of an AFO with mechanical actuators is lower than that of an AFO with electrical actuators.

**The effects of an AFO with gait control on the user** in the short- and long-term have not yet been fully investigated. In this paper, the effect of the AFO is defined as the improvement of the user kinetics (in this case without an AFO) after the AFO has been provided for a certain period (short-term and long-term). For example, a patient used an AFO for two weeks. Then, on the third week, the user gait without the assistance of the AFO was analyzed. The effects of the provision of an AFO on rehabilitation was measured on an AFO with a bending stiffness control [57] in the short-term, long-term, and even in terms of the immediate effect (during the provision of the AFO) [16,19]. The immediate effects of an AFO with gait control, such as damping stiffness, assistive torque, and motion control, on the user have been reported, such as a higher walking speed, longer stride length, and so on [72], but without focusing on the short-term or long-term utilization [74,96]. Another work discussed the long-term effect of assistance by an AFO by measuring the user kinetics and the emotional improvement of the patient [60]. Moreover, the participants in the AFO experiment were mostly healthy subjects, if not younger patients with some disabilities. Therefore, the lack of participation by the elderly in the AFO experiment should be addressed in a future assessment of the AFO effects [97].

Although AFO with robotic technology is the most sophisticated system compared to other AFO versions, such as the flexible rigid AFO and the articulated AFO with a mechanical actuator, conventional AFO has not been fully investigated, especially with regard to its short-term and long-term effects. Such studies should be continued, because each of the developed AFOs serve a different purpose, depending on the rehabilitation need, such as for walking or running, and so on. It will be interesting to develop an AFO that can serve many purposes with simple adjustments.

## 6. Future Research

The following are the research possibilities for the development of the AFO, especially with regard to the control of its mechanical properties:

(1)    Adaptive output reference for damping stiffness control

As explained earlier, the mechanical properties of the AFO should match the needs of users, to maximize the benefits of the AFO [98,99]. The AFO with damping stiffness control already has advantages compared to the others (bending stiffness, assistive torque, and motion control) such as optimized weight, cost, and safety, because the actuator is a passive actuator. Despite that, the damping stiffness control has not been optimized when using an adaptive output reference. The damping stiffness has been adapted to different gait phases; however, it has not yet been adapted to different persons and environments. The adaptive output reference has been conducted using the GRF for knee damping stiffness, but not yet for ankle damping stiffness. Thus, it would be interesting to apply the adaptive output reference to the ankle damping stiffness control.

(2)    AFO gait control performance analysis

Most of the performance analyses of the gait control of the AFO were conducted by comparing the gait kinematics between barefoot walking and walking with the developed AFO. In some cases, the AFO properties were changed to analyze the user kinematics, the walking energy cost, and the device energy cost before and after walking. However, the analysis of the gait control performance has not been reported in existing works, except through a literature review in which the vast variety of procedures in each reported works has made it difficult for a comparative study to be done [100]. Therefore, it is proposed that an analysis of the existing works should be carried out, using the same procedure, to determine the most preferable gait control strategy.

(3)    The effects of an AFO with the provision of gait control

The immediate effects of using an AFO with gait control have been presented, but not the short-term and long-term effects. It will be interesting to determine the effects of an AFO with robotic assistance and gait control among patients in the short-term and long-term as one of the aspects for analyzing the performance of the AFO with regard to gait control. The inclusion of more elderly participants in the AFO experiment should also be considered.

## 7. Conclusions

This paper discussed the works on a gait control strategy for AFOs, especially with regard to the control of the mechanical properties, such as the control of the bending stiffness, damping stiffness, assistive torque, and motion path. The control of the mechanical properties in previous works has been reviewed in terms of the input and number of gait phases, output reference estimation, and performance evaluation. From the review that was made, it is suggested that the adapted output reference method be used for the damping stiffness control so as to maximize the benefits of the AFO to the user. Also, with regard to the gait control performance analysis, a comparison should be considered between the existing gait control and the user's before–after condition. Finally, the short-term and long-term effects of the provision of gait control to the user should also be investigated further.

**Funding:** The research received no extra funding.

**Acknowledgments:** The authors would like to express their appreciation to the Ministry of Education and Universiti Teknologi Malaysia for their support under the research university grant [VOTE: 19H94]. To Universitas Sebelas Maret for their support in product development of MR brake for Ankle Foot Orthoses under Hibah Kolaborasi Internasional 2019. Also, to Institut Teknologi Telkom Surabaya for their support in publication under Dana Publikasi Jurnal Ilmiah ITTelkom Surabaya.2019 The funders had no role in the study design, data collection and analysis, decision to publish, or preparation of the manuscript.

**Conflicts of Interest:** The authors declare no conflicts of interest.

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
