# Peer review of "A Review on the Control of the Mechanical Properties of Ankle Foot Orthosis for Gait Assistance"

_actuators, doi:10.3390/act8010010_

Round 1

Reviewer 1 Report

The paper presents a review of approaches to the problem of mechanical properties control for ankle foot orthosis (AFO) in gait assistance of post-stroke patients. Since it is a review work, it attempts to offer a systematic overview of research in the particular field so that it can help other researchers in designing their own AFOs based on their application. The topic is very specific but certainly is of interest of researchers and/or practitioners in the field of rehabilitation engineering. In this case the list of references might be of special importance and authors use mostly very recent works which is a positive factor to maintain up-to-date character of the work. Even though the work itself is quite extensive, it is relatively common to use higher number of references in review articles (100 and more) – it might be difficult to set objective threshold for this but higher number of works naturally offers a more general picture. I have some minor remarks which authors could address in their article:

1.       I definitely recommend having the paper re-checked by native speaker. Even though it does not contain extremely serious language errors, it is not error-free and in certain places they obscure the actual meaning.

2.       I suggest reconsidering the hierarchy of numbering in distinctive parts of the paper to make it more tidy – e.g. it appears that paragraphs 5,6 and 7 (and possibly 8) are part of a topic dealing with output reference estimation and control.

3.       Figure 12 is, I suppose, based on work of other researchers so it should be properly referenced – the number of reference is missing there.

4.       From general point of view, what I miss most is maybe more specific description of criteria used for choosing particular research work to be included into review. This remark might be related to the above mentioned total number of references – whether more works could be used but did not match the criteria for being selected. This question, for instance, arises in the case of used actuators (other types than those mentioned) or actual control method (e.g. adaptive vs. learning?).

To conclude, the paper is interesting and certainly will be of value to those working in the field of biomedical engineering. On the other hand, it certainly needs some modifications and/or corrections to further improve its quality.

Author Response

Referee 1

Comments:

The paper presents a review of approaches to the problem of mechanical properties control for ankle foot orthosis (AFO) in gait assistance of post-stroke patients. Since it is a review work, it attempts to offer a systematic overview of research in the particular field so that it can help other researchers in designing their own AFOs based on their application. The topic is very specific but certainly is of interest of researchers and/or practitioners in the field of rehabilitation engineering. In this case the list of references might be of special importance and authors use mostly very recent works which is a positive factor to maintain up-to-date character of the work. Even though the work itself is quite extensive, it is relatively common to use higher number of references in review articles (100 and more) – it might be difficult to set objective threshold for this but higher number of works naturally offers a more general picture.

Suggestions:

I have some minor remarks which authors could address in their article:

1.      I definitely recommend having the paper re-checked by native speaker. Even though it does not contain extremely serious language errors, it is not error-free and in certain places they obscure the actual meaning.

Answer: Thank you for the comment. The language has been checked by native editor. I hope it improved the manuscript as a whole.

2.      I suggest reconsidering the hierarchy of numbering in distinctive parts of the paper to make it more tidy – e.g. it appears that paragraphs 5,6 and 7 (and possibly 8) are part of a topic dealing with output reference estimation and control.

Answer:

Thank you for the comment. The numbering of the section in the paper had been reconsidered such as:

1.      Introduction

2.      AFO structure types

3.      Input and gait phases

4.      Output reference estimation and control

4.1  Bending stiffness control

4.2  Damping stiffness control

4.3  Assistive torque

4.4  Motion path control

4.5   

5.      Discussion

4.1  Input consideration for gait phase classification

4.2  Output reference: fix versus adaptive

4.3  Controller performance evaluation

6.      Future research

7.      Conclusion

The numbering part has been highlighted in the paper.

3.      Figure 12 is, I suppose, based on work of other researchers so it should be properly referenced – the number of reference is missing there.

Answer: Thank you for the comment, the reference number has been added to the Figure 12.

4.      From general point of view, what I miss most is maybe more specific description of criteria used for choosing particular research work to be included into review. This remark might be related to the above mentioned total number of references – whether more works could be used but did not match the criteria for being selected. This question, for instance, arises in the case of used actuators (other types than those mentioned) or actual control method (e.g. adaptive vs. learning?).

Answer: Thank you for the comment.

The paper is focused on the AFO development for post-stroke rehabilitation rather than lower-limb assistive device which may include exoskeleton, Knee AFO (KAFO), and prosthesis. The lower-limb assistive devices able to assist the patient who unable to walk at all. On the other hand, the AFO only partially assist the patient with simpler configuration than lower-limb assistive devices. Therefore, for less disability people such as post-stroke patient with foot drop, it is more suitable to use the AFO which highlighted in this paper. Note that several works regarding the lower-limb assistive devices have been included to get some idea to develop the AFO.

However, the number of citation which corelated to the recent works on AFO have been increased and the information of criteria used for choosing particular research work have been added in the introduction section.

To conclude, the paper is interesting and certainly will be of value to those working in the field of biomedical engineering. On the other hand, it certainly needs some modifications and/or corrections to further improve its quality.

Closing:

We have sent the manuscript to native proofreader to improve the scientific rigor and corrected grammatical errors throughout the revised manuscript. We appreciate all the constructive comment which enrich the discussion in this review manuscript. If you have any question about this paper, please kindly contact us as soon as possible. Once again, thank you for all your help. We are looking forward to hearing your reply soon.

Best regards,

Sincerely yours

Dr. Mohd Azizi Abdul Rahman

Reviewer 2 Report

This manuscript provides a systematic review of the Ankle Foot Orthosis devices with an emphasis on the joint flexibility, control input, output, and performance evaluation. This paper is very informative, and the importance of the study is well explained. However, besides the technical details, the high-level improvement and progress in the use of the AFO devices for rehabilitation should also be discussed in order attract the interests of physicians in addition to engineers. Besides, several issues need to be well addressed before publication:

·         It is needed to include physical pictures of several representative AFO devices so that readers can have an idea of how those devices look like.

·         Line 75 enhanced with robotic should be “enhanced with a robot” or “enhanced with robotic technology”;

·         The last paragraph on Page 2 should include relevant literature to provide an overview of the technology evolvement.

·         Line 87-88: The first sentence needs to be rewritten. “… can be categorized into two groups…”

·         Line 95-96: “Both control strategies have their own pros and cons…” The authors are suggested to provide more details about what those “pros and cons” are.

·         In section 2, the AFO structure types are not well explained. It will be better if the authors can provide some exemplary or representative work.

·         The size of table 1 should be carefully set to allow all words to be displayed properly.

·         Since the controller performance and evaluation results are important as discussed in section 12, it is necessary to include the performance results in Table 1. So that readers will be able to have an overview of how well the reviewed APO devices performed.

Author Response

Answers to Reviewer’s Comments

Manuscript ID: actuators-421139

Title: Review on Ankle Foot Orthosis Mechanical Properties Control for Gait Assistance

Thank you very much for the valuable and very detailed comments. We tried our best to accommodate all of the comments and recommendations in the revised version. We sincerely feel that the comments have made the revised manuscript reflect much better than what was intended in this work.

Please note that the changed parts are highlighted in the revised version.

REFEREE REPORT(S):

Referee 2

Comments:

This manuscript provides a systematic review of the Ankle Foot Orthosis devices with an emphasis on the joint flexibility, control input, output, and performance evaluation. This paper is very informative, and the importance of the study is well explained. However, besides the technical details, the high-level improvement and progress in the use of the AFO devices for rehabilitation should also be discussed in order attract the interests of physicians in addition to engineers.

Answer: Thank you for the comment. The explanations about the improvement and progress in the use of the AFO devices for rehabilitation have been added in the discussion section.

Suggestions:

Besides, several issues need to be well addressed before publication:

1.      It is needed to include physical pictures of several representative AFO devices so that readers can have an idea of how those devices look like.

Answer: Thank you for the comment. Before this, the illustration is used because the permission to reuse from the other work has not been granted from the publisher. The Figure that already has the permission to reuse has been changed in the manuscript such as figure 6, 7, and 10.

2.      Line 75 enhanced with robotic should be “enhanced with a robot” or “enhanced with robotic technology”;

Answer: Thank you for the comment. The word has been changed as per requested.

3.      The last paragraph on Page 2 should include relevant literature to provide an overview of the technology evolvement.

Answer: Thank you for the comment. Relevant literature review has been provided in the manuscript.

4.      Line 87-88: The first sentence needs to be rewritten. “… can be categorized into two groups…”

Answer: Thank you for the comment. The first sentence has been rewritten into “Based on the controller output reference estimation method, the control strategies can be categorized into two groups. They are subject-oriented Figure 2 (a) and phase-oriented control types (Figure 2 (b).”

5.      Line 95-96: “Both control strategies have their own pros and cons…” The authors are suggested to provide more details about what those “pros and cons” are.

Answer: Thank you for the comment. More details about pros and cons of both control strategies have been provided in the manuscript such as “The subject-oriented control has more complex, but one calculation only because it must consider the whole walking gait. On the other hand, the phase-oriented control has more calculation due to the number of gait phases, however with relatively simpler calculation.”

6.      In section 2, the AFO structure types are not well explained. It will be better if the authors can provide some exemplary or representative work.

Answer: Thank you for the comment. The AFO structure types explanations has been added as well as the illustration of the AFO structure types.

7.      The size of table 1 should be carefully set to allow all words to be displayed properly.

Answer: Thank you for the comment. The table size has been increased to allow all words to be displayed properly. Consequently, the table was made landscape instead of portrait.

8.      Since the controller performance and evaluation results are important as discussed in section 12, it is necessary to include the performance results in Table 1. So that readers will be able to have an overview of how well the reviewed APO devices performed.

Answer: Thank you for the comment. The information about the performance result has been included in Table 1.

Closing:

We have sent the manuscript to native proofreader to improve the scientific rigor and corrected grammatical errors throughout the revised manuscript. We appreciate all the constructive comment which enrich the discussion in this review manuscript. If you have any question about this paper, please kindly contact us as soon as possible. Once again, thank you for all your help. We are looking forward to hearing your reply soon.

Best regards,

Sincerely yours

Dr. Mohd Azizi Abdul Rahman

Round 2

Reviewer 2 Report

All my previous concerns are well addressed. Only need to further proofread and improve the writing style.